# Effect of Treadmill Training with Visual Biofeedback on Selected Gait Parameters in Subacute Hemiparetic Stroke Patients

**DOI:** 10.3390/ijerph192416925

**Published:** 2022-12-16

**Authors:** Katarzyna Kaźmierczak, Agnieszka Wareńczak-Pawlicka, Margaret Miedzyblocki, Przemysław Lisiński

**Affiliations:** Department and Clinic for Rehabilitation and Physiotherapy, University of Medical Sciences, 28 Czerwca 1956 Str., No 135/147, 60-545 Poznań, Poland

**Keywords:** sub-acute stroke, neurologic locomotion disorder, neurorehabilitation, treadmill training, biofeedback, hemiparetic gait

## Abstract

Background: Functional limitations after a stroke are unique to each person and often include impaired independent mobility. A reduction in existing gait deficits after a stroke is often one of the main goals of rehabilitation. Gait re-education after stroke is a complex process, which consists of the effects of many therapeutic interventions. Objective: The study aimed to analyze the effects of using a treadmill with visual feedback in gait re-education in the sub-acute stroke period and assess the impact of biofeedback treadmill training on selected gait parameters, improving static balance and reducing the need for orthopedic aids. Methods: The study included 92 patients (F: 45, M: 47) aged 63 ± 12 years, with post-ischemic sub-acute (within six months onset) stroke hemiparesis, treated at a neurological rehabilitation ward. All patients participated in a specific rehabilitation program, and in addition, patients in the study group (*n* = 62) have a further 10 min of treadmill training with visual feedback. Patients in the control group (*n* = 30) participated in additional conventional gait training under the direct supervision of a physiotherapist. The evaluation of static balance was assessed with the Romberg Test. A Biodex Gait Trainer 3 treadmill with biofeedback function was used to evaluate selected gait parameters (walking speed, step length, % limb loading, and traveled distance). The use of an orthopedic aid (walker or a crutch) was noted. Results: After four weeks of rehabilitation, step length, walking speed, traveled distance, and static balance were significantly improved for the study and control group (*p* < 0.05). Treadmill gait training yielded significantly better results than a conventional rehabilitation program. Only the study group observed a corrected walking base (*p* < 0.001). All participants showed a reduction in the use of walking aids (*p* = 0.006). There was no asymmetry in the % of limb loading for either group prior to or following rehabilitation. Conclusions: The treadmill with visual biofeedback as conventional gait training has resulted in a significant improvement in parameters such as step length, walking speed, static balance, and a reduction in the use of locomotion aids. However, the achieved improvement in gait parameters is still not in line with the physiological norm.

## 1. Introduction

Current publications present consistent research results confirming the occurrence of disturbances in some characteristic gait parameters in people after stroke. There are undoubtedly important parameters, such as reduced walking speed, step length, time on the injured side, and asymmetric gait patterns [1,2,3].

Stroke survivors typically have a decreased stance phase and a prolonged swing phase of the paresis limb. Further, it results in a shorter stride length and a slower walking speed [4]. The range of step lengths of the normative values is quite broad. This variable becomes a new value depending on age or sex. Oatis [5] assumes that in a healthy adult, the average stride length is 0.7 to 0.81 m. Previous studies have also shown that the intermediate step length in people after stroke is 0.36 to 0.6 m [2,6,7]. Undoubtedly, the improvement of gait is aimed at improving the symmetry of step length, which means the similar step length of both limbs

Gait speed is another commonly used indicator of locomotor efficiency in healthy individuals. Normative values for a healthy fast-walking adult range between 0.82 and 1.62 m/s [8]. The hemiparetic gait of post-stroke patients is characterized by a significant reduction in gait speed [9,10]. According to the publication, the walking speed of people with hemiplegia often varies between 0.38 m/s in the subacute phase [11] to 0.5 m/s in chronic stroke [12]. It results from insufficient compensation of the motor deficits characteristic of people with hemiparesis [13,14,15].

Balance disturbances are also observed in this group of patients. The primary indicator of good balance during gait is the symmetry of the load on the lower limb (50% of the load on the right and left limb). Patients with post-stroke hemiparesis demonstrate an incomplete load on the paretic lower limb [16,17,18]. According to many authors, balance is influenced by the symptoms characteristic of strokes, such as impaired proprioception, hypertonicity, and reduced strength of the muscles of the paresis limb. All these variables cause its underload when standing and walking [9,19,20,21].

The improvement of gait is an essential element of functional rehabilitation after stroke and the primary therapeutic goal for patients [22]. In clinical practice, gait recovery training in post-stroke patients is often distracted in two phases. The first includes muscle-strengthening exercises [23,24,25] and neuromuscular re-education (PNF, NDT-Bobath), which increases ROM (range of motion) by increasing muscle length and improves neuromuscular performance [26,27,28,29,30]. Then, in the second phase, the emphasis is on the practical use of the results achieved in the first phase. Here, training will be focused on eliminating abnormal gait patterns, practical gait training, and improving the body’s efficiency.

Recently, this second-phase gait training has been extensively analyzed in the literature. The publications have presented the results of studies comparing the effectiveness of treadmill walking training with the conventional over-ground walking training program in patients after stroke [31,32], and the effectiveness of the body weight-supported treadmill training [33,34,35].

In modern gait re-education, it is desirable to use tools that allow the patient to control the correctness of the activity performed. A new tool in gait re-education in stroke patients is the combination of treadmill gait training with visual feedback. This method is gaining popularity in clinical practice [36] During such gait training, the patient is informed about the need to correct the performance of the task performed by an external visual stimulus, activated when the patient moves beyond the desired range [37]. Only a few types of research have been dedicated to evaluating whether visual biofeedback plays an essential role in stroke recovery and may be a valuable approach to stroke rehabilitation [1,9,10,11,38,39].

Therefore, more research is needed to determine the effect of biofeedback on gait therapy.

The study aimed to analyze the effects of a treadmill with a visual feedback rehabilitation program in early post-stroke physiological gait re-education. Our primary outcome was to evaluate changes in gait parameters such as step length and limb loading on the paretic limb. Secondly, we focused on assessing other selected gait parameters, such as walking speed, sense of balance, or the need for orthopedic aids to walk.

We have formulated the following research areas:Effects of the treadmill with visual feedback on the step length and load on paretic and healthy limbs.Effects of the treadmill with visual feedback on walking speed.Effects of the treadmill with visual feedback on improved functional gait variables, such as sense of balance and walking base, and decreasing the need for orthopedic aids.

## 2. Materials and Methods

A trial of post-stroke hemiparesis patients was carried out between June 2018 and March 2020 at the Neurological Rehabilitation Ward of Wiktor Dega Orthopedic-Rehabilitation Clinical Hospital, Poznań University of Medical Sciences.

The inclusion criteria were: ischemic stroke episode within six months after onset, ability to walk independently with or without orthopedic aid, functional disability score in the Barthel Index > 10 points, hypertonicity of the paretic lower limb (Ashworth ≤ 1 plus), and paretic ankle dorsal flexors muscle strength ≥ 2 (MRC scale, Medical Research Council scale). The minimum treadmill walking speed was 0.14 m/s (the lowest velocity necessary for starting the visual biofeedback function). Therefore, the respondents had to obtain a result of ≥71.4 s in 10 MWT (Minute Walk Test).

Each study participant had a diagnosis of ischemic stroke confirmed by a neurologist and neuroimaging study (computed tomography-CT or magnetic resonance imaging—MRI). Patients with the following conditions were excluded from the study:

Cognitive functional impairment impeding understanding of commands (AMTS, Abbreviated Mental Test Score), previous stroke episodes, movement disorders significantly limiting task performance, coexisting orthopedic disorders (including primary hip or knee arthroplasty, lower limb amputation at any level, fixed joint contractures) and rheumatoid conditions affecting ambulation.

Cognitive abilities were determined by our clinical psychologist using the AMTS, in which the patient had to obtain a score of at least 9 points to qualify for the study. The research methodology required an appropriate level of cognitive performance and concentration, hence the decision to use the AMTS as a qualifying test for exclusion from the study [40]. Of 171 post-stroke sub-acute (up to 6 months after onset) hemiparetic patients, 92 qualified for the study.

The study participants were divided into two groups. Both patients from the study group (*n* = 62) and those from the control group (*n* = 30) participated in a specific rehabilitation program used in our clinic. The use of treadmill gait training with visual feedback in patients with subacute stroke is a new therapeutic tool. The study group is double the size of the control group because the therapy results are already known in the literature [41]. The study participants then underwent additional gait training. The difference in the rehabilitation programs between the groups concerned only the methods used in gait rehabilitation. The patients in the study group have an additional 10 min of treadmill training with visual feedback to relearn their gait. The patients in the control group participated in conventional gait training under the direct supervision of a physiotherapist. This gait training consisted of moving on flat ground with the necessary orthoses or orthopedic aids. During this gait re-education, attention was naturally paid to, among others, the symmetry of steps, body posture in motion, and a gradual increase in speed and distance.

The gait parameters of each participant in the control and study groups were measured twice in the treadmill test. The first measurement occurred before the start of the treatment and again on the last day of treatment, after four weeks. Data such as step length, walking speed, limb load, and distance traveled were measured on the treadmill and shown on the monitor after gait training on the Biodex Gait Trainer 3. A physiotherapeutic examination assessed other functional gait variables. 

The patients in the study group averaged 63.0 ± 10.9 years of age (27.0–85.0), while the patients in the control group averaged 61.9 ± 14.0 years of age (30.0–87.0). The study group consisted of 29 women (46.8%) and 33 men (53.2%), with 34 left-sided stroke patients (54.8%) and 28 right-sided stroke patients (45.2%). The control group comprised 16 women (53.3%) and 14 men (46.7%). In this group, 18 patients (60%) had a left-sided stroke, and 12 (40%) had a right-sided stroke. No significant differences in age (*p* = 0.816), height (*p* = 0.705), weight (*p* = 0.649), or BMI (*p* = 0.708) were found between the control and study groups. No statistical differences were observed. The characteristics of the study population are shown in Table 1.

### 2.1. Procedure and Instruments

Medical documentation and clinical patient evaluation were used to qualify individuals for the study. Tibialis anterior muscle strength was determined using the Medical Research Council’s Scale (MRC scale), while the Modified Ashworth Scale was applied to assess the spasticity of the paretic muscles [42]. The patient was required to cover a distance of 10 m (10 Meters Walk Test), and the time to complete the task was measured. During this test, the subject could, if required, use an orthopedic aid such as a walker or a crutch. The walking base was assessed visually. The subjects’ walking floor was considered normal when it approximated hip width; however, when the distance between the ankles was noticeably shorter or longer than hip-width, the walking base was considered abnormal. The patient’s degree of independence was determined using the Barthel Index, and balance was assessed with the Romberg test [43]. The Romberg test was carried out 3 times, and the given result is the average value.

The Romberg test is used to diagnose gait disorders caused by incorrect proprioception with information about the location of the joints. It is a sensitive tool that accurately measures the degree of imbalance caused by central and peripheral vertigo and head trauma [6,26]. Clinically, the Romberg test has been used for 150 years to evaluate patients with impaired balance due to sensory and motor disorders [26]. In the Romberg test, the patient stands upright and is asked to close his eyes.

The test result is positive when the patient cannot maintain balance with closed eyes. Losing balance can be understood as increased body sway, one foot in the direction of the fall, or even a fall [2,7]. The examiner now observes any imbalances for one minute. Loss of balance can be defined as increased body swaying and movement of the foot towards the falling or falling.

A Biodex Gait Trainer 3 (BIODEX MEDICAL SYSTEMS, INC., New York, NY, USA) treadmill with real-time step length visualization (where the foot was placed) was used to assess the selected spatiotemporal gait parameters. During the test, the patient was visually informed about the suggested foot placement/planned length of the step. This three-minute extended test was conducted before and after the rehabilitation program. The following parameters were recorded during each trial:step length for the paretic and the healthy limb (cm)load on the paretic and the healthy limb (%)walking speed (km/h)

The average values for each parameter were calculated for further analysis.

### 2.2. Protocol

While hospitalized at the neurological rehabilitation ward, patients from the study and control groups participated, six days per week, in a personalized rehabilitation program used at the clinic. In the first step, therapy was based on individual exercise sessions with a therapist (60 min in length). The program described the first phase of rehabilitation in our clinic. The exercises performed here are designed to improve muscle strength, coordination, task-specific training to improve walking ability, balance (sitting and standing), and activities that involve neurophysiological methods (PNF).

In the second phase of a daily rehabilitation program, following a rest period of a minimum of 30 min, the patients underwent an additional 10 min of training focused on improving selected gait parameters. For the participants of the study group, there was training on a treadmill with visual biofeedback, while the control group additionally participated in individual gait training conducted by a physiotherapist.

The Biodex Gait Trainer 3 treadmill is equipped with load sensors and software showing the weight distribution transferred to each foot during the support phase. Information on the actual foot location was presented to patients graphically (foot contours) on the screen. During treadmill training, the patient was required to adjust their step length to the range displayed on the monitor by two parallel lines. The patient was continuously informed, in a graphic manner, about the results achieved, including the step length and expected location of foot placement. If the lines determining the correct step length were crossed, the patient was alerted with a sound signal. When the difference between the expected and actual step length was significant, text information prompting a corrective measure (e.g., “right step longer”) appeared on the screen.

Training walking speed at the beginning of the four-week program was selected based on the inclusion criteria of walking speed (min 0.14 m/s). With the improvement of gait quality, determined by values of tested gait parameters, the therapist systematically increased the pace and duration of the training, adjusting the settings to the current fitness level of the examined patient.

### 2.3. Statistical Analysis

Data were analyzed with the Statistica™ version 13.1. Demographic data and clinical characteristics are presented as means, standard deviations (SD), median, and range. The Shapiro–Wilk test was used to assess the normality of distributions in the test scores. Independent *t*-tests were performed to compare the general participants’ characteristics. Repeated-measures analysis of variance was conducted to assess the group-by-time interaction effect of interest. When a significant interaction was found, Fisher’s LSD test was used to compare the outcome variables between and after the intervention in each group. Categorical variables were presented as absolute numbers and relative frequencies. McNemar’s test was used to compare dependent proportions. The *p*-values of less than 0.05 were considered statistically significant.

## 3. Results

Table 2 and Table 3 show the results obtained by the study and control groups during the treadmill gait training before and after four weeks of rehabilitation. Before therapy, no significant differences in examined gait parameters (paretic and healthy limb step length, percentage of loading of each limb, walking speed, and distance traveled) were found between the groups. Except for the limb loading parameter, all other pre-rehabilitation parameters were below the physiological norm.

Step length measurements before rehabilitation were compared to measurements following rehabilitation (Table 2). The main effect of the groups was significant (*p* < 0.001). There was also a significant interaction in the mean step length scores between the groups and the time (*p* = 0.018). Both therapies, the conventional program used in the control group and the treadmill gait training combined with the standard program used in the study group, improved step length. In the study group, the step length of the paretic limb increased by 9.8 ± 9.9 cm (*p* < 0.001), while in the control group by 3.4 ± 5.5 cm (*p* = 0.023). For a healthy limb, the results improved by 10.4 ± 10.5 cm in the study group (*p* < 0.001) and by 4.3 ± 6.7 cm in the control group (*p* = 0.005). The results after therapy differed significantly between healthy limbs in the groups, which may suggest that treadmill gait training yielded significantly better results than conventional rehabilitation programs for the healthy limb (*p* = 0.04). No significant differences in the load of the limb between groups were observed. Graphical changes in step length are presented in Figure 1.

Both groups’ walking speeds and distances traveled (Table 3) significantly improved after therapy. A comparison of the therapy results between both groups was performed. The walking speed and distance traveled results after therapy differed significantly between the groups. In the study group, the walking speed increased by 0.19 ± 0.16 m/s, while in the control group by 0.08 ± 0.09 m/s. The distance traveled improved by 35.4 ± 38.7 m in the study group and by 14.4 ± 16.9 m in the control group. The results showed significantly more improvement in the study group for walking speed (*p* = 0.023) and distance traveled (*p* < 0.001). Changes in walking speed and distance are illustrated in Figure 2 and Figure 3.

Additionally, the Romberg Test-assessed body balance significantly improved for patients in both the study (*p* < 0.001) and the control (*p* = 0.006) groups. Although gait quality improved in both groups, only the study group participants showed a statistically significant (*p* < 0.001) walking base refinement. Furthermore, the number of patients requiring a walking aid significantly decreased (*p* = 0.006) in both groups (Table 4).

## 4. Discussion

Only a handful of scientific reports assess the therapeutic effectiveness of the qualitative and quantitative improvement of the chosen gait parameters after rehabilitation in the early stage post-stroke. Most studies evaluate these gait parameters (step length, walking speed, % limb load, and traveled distance) in stroke survivors at later stages (>6 months post-stroke) [9,10,22,44]. In this last period, some abnormal movement patterns have already been established, and restoring physiological gait patterns is difficult. Therefore, it seems advisable to introduce early gait re-education to prevent the consolidation of pathological gait patterns and unfavorable compensations. Undoubtedly, rehabilitation should begin as soon as possible after a stroke. It should be goal-oriented therapy relevant to the patient’s needs, mobility, and walking [45].

The results of our research indicate improvement of some spatiotemporal gait parameters in patients in the early stage of stroke undergoing rehabilitation using biofeedback. After four weeks of gait training with biofeedback, we observed an improvement in step length (Table 2). The average walking speed also improved in this four-week period, which naturally led to lengthening the distance covered (Table 3). Therapeutic efficacy also manifested in improved body balance, reduced the requirement for walking aids, and decreased the number of patients with an abnormal walking base (Table 4). The results of our study show the beneficial effect of the treatment and guide the proposed model of biofeedback walking therapy in people with early stroke (< 6 months onset). The introduction of treadmill walking with biofeedback as a procedure used in gait rehabilitation after stroke is a practical solution for stroke therapy.

### 4.1. Step Length

Until today, several researchers reported improvements in step length and symmetry due to treadmill gait therapy [10,33,46]. Contrary to these findings, Patterson et al. [47] did not observe a significant improvement in step symmetry for the early post-stroke period,

Our study reflects the research mentioned above results on improving stride length (Table 2). We obtained an improvement in stride length in each of the research groups. The participants’ training gaits with biofeedback significantly improved their step length on paretic limbs (*p* < 0.001), similar to the improvement observed in the control group with additional gait training with a physiotherapist (*p* = 0.023). However, these results were still insufficient to approach the norm of healthy people (normative step length on average 0.72 m) [48]. After four weeks of rehabilitation, the participants of our study still had an average step shorter by 0.2 m compared to the norm of healthy people.

### 4.2. Lower Limb Loading

The asymmetrical nature of the gait with hemiparesis is well documented. One of the observed gait variables is the load on the lower limb with hemiplegia being reduced compared to the non-paralyzed side [17,49,50].

Many studies assess the asymmetry of the loading of the lower limbs while walking in people with chronic stroke [8,51,52]. One of the directions represented in the field of gait re-education in chronic stroke focuses on reducing the activity of the non-paralytic limb during gait exercises. One of the therapeutic solutions equalizing the load on the lower limbs while walking was proposed by Bonnyaud et al. [52]. These researchers place an additional load on a non-paretic limb during gait rehabilitation. In the available literature, we found only one work that exposed the asymmetry of lower limb loading during gait in people up to 6 months after a stroke. In an average of 3 months after stroke, stroke survivors were the subject of studies by Ribeiro et al. [53]. Those researchers added balance exercises to the gait re-education program. When walking on a treadmill, study participants wore an additional weight of 5% of their body weight strapped to the ankle limb without paresis. The results of this study suggest that treadmill-assisted gait training with an extra load on the stem without paresis, combined with weight transfer and balance exercises, may be effective in reducing the asymmetry of the limb load during walking.

Our data analysis did not show any significant asymmetry of lower limb load in our study participants. However, it is worth noting that the gait re-education process in our research started as soon as possible after the patient left the neurological ward. Therefore, the mechanisms of the asymmetric loading of the lower limbs during gait probably did not develop. Both before and after the end of the four-week rehabilitation program, the load on the stem with and without paresis was approximately 50% on average (Table 2). We have implemented comprehensive rehabilitation, including balance exercises in the first rehabilitation phase and then in the second phase, gait re-education early (<6 months) after a stroke. Therefore, it is possible that the introduction of gait exercises with biofeedback and balance exercises in the early period after stroke onset resulted in no significant difference in lower limb loading during gait after the 4-week rehabilitation period. It is also possible that this early intervention minimized the excessive strain on the limbs while walking, seen in post-stroke patients at a later stage [1,9]. However, more research is needed to include larger groups of patients at different stages of stroke.

### 4.3. Speed

According to the standards cited by Murtagh et al. [54], the standard walking speed of healthy people ranges from 0.82–1.62 m/s. Similar values are presented by Whittle et al. [39]; for women aged 50–64 years, the required walking speed is 0.91–1.63 m/s, and for men at this age, it is slightly higher (0.96–1.68 m/s). Many stroke survivors recover the ability to walk during early rehabilitation, but most still present a slower walking speed. Therefore, improving this gait parameter is one of the goals of rehabilitating people after stroke [16,55,56,57,58]. Our research shows a significant improvement in walking speed after four weeks of rehabilitation both in the research group (biofeedback treadmill training mean from 0.47 to 0.66 m/s) and the control group (conventional gait rehabilitation mean from 0.49 to 0.56 m/s). The factor that may affect the improvement of walking speed is balance. Exercises used in our rehabilitation program to strengthen the muscles of the lower limbs, exercises for the motor control of the paresis of the lower limb, and training for postural, static, and dynamic balance undoubtedly improve walking speed for people after a stroke [34,59,60]. However, these values differ from the norm presented by healthy people. Probably four weeks of rehabilitation is too short a period to receive a walking speed commonly accepted for healthy people.

### 4.4. Balance

Good body balance determines walking efficiency, progressive movement flow, and safety when changing directions [61]. Therapy used in our study led to significant balance improvement, evidenced by a significant (*p* < 0.001) change in Romberg test results and decreased use of walking aids (walker, crutch) in the study group patients (Table 4). Comparably, Juan Li et al. [62] also showed improvement in post-stroke balance and mobility using rehabilitation methods similar to those used in our study.

Visual biofeedback plays an essential role in improving body balance and, consequently, gait efficiency [63,64]. However, these are often studies involving popular balance training on a stabilometric platform [65,66]. No one has analyzed the walking base in the research so far. Increasing the walking base by spacing the lower extremities apart, broader than hip-width, would provide patients with more outstanding balance while walking. Our research also observed the decreased use of walking aids (crutches, walkers, etc.). Gait training on the treadmill with biofeedback also positively affected the walking base, causing its significant (*p* < 0.001) reduction. In the control group, no such effect was observed (*p* = 0.074).

### 4.5. Summary

The applied rehabilitation algorithm improved the gait parameters such as step length, walking speed, and distance covered after the four-week rehabilitation program in both the study and the control group. However, an additional analysis comparing the outcomes of the two therapeutic methods showed that standard rehabilitation supplemented by a treadmill with visual biofeedback training significantly improved these parameters more than standard rehabilitation alone (see Table 2 and Table 3).

### 4.6. Limitations

Using only one relatively simple method to assess gait improvement can also be considered a limiting factor. There is no doubt that computer posturography is more sensitive than the clinical Romberg test in detecting postural control disorders, and this measurement should be evaluated to obtain more reliable results of postural stability. A potential limitation of this study was also the sample size, so it can be considered a pilot study. In the future, we recommend research on a larger group to observe differences in the effectiveness of treadmill training with visuals. A more reliable evaluation of the visual effects of biofeedback in re-education would be to compare the results of more therapeutic groups, e.g., groups on the treadmill with feedback and without feedback. This is the subject of further research. Additionally, the short rehabilitation time (four weeks) may not be sufficient to determine more significant differences between the groups. We also did not address how long the effects persisted. A future study that includes the measurement of long-term outcomes is advisable.

## 5. Conclusions

Early introduced rehabilitation in sub-acute stroke combined with additional gait rehabilitation provides significant improvement in step length, walking speed, sense of balance, and walking base. Gait rehabilitation with visual biofeedback treadmill training is a more effective method of gait re-education in subacute post-stroke patients than rehabilitation training with typical gait re-education. The proposed gait rehabilitation with visual biofeedback treadmill training in the subacute stroke period significantly contributes to the sense of body balance improvement and reduces using walking aids.

## Figures and Tables

**Figure 1 ijerph-19-16925-f001:**
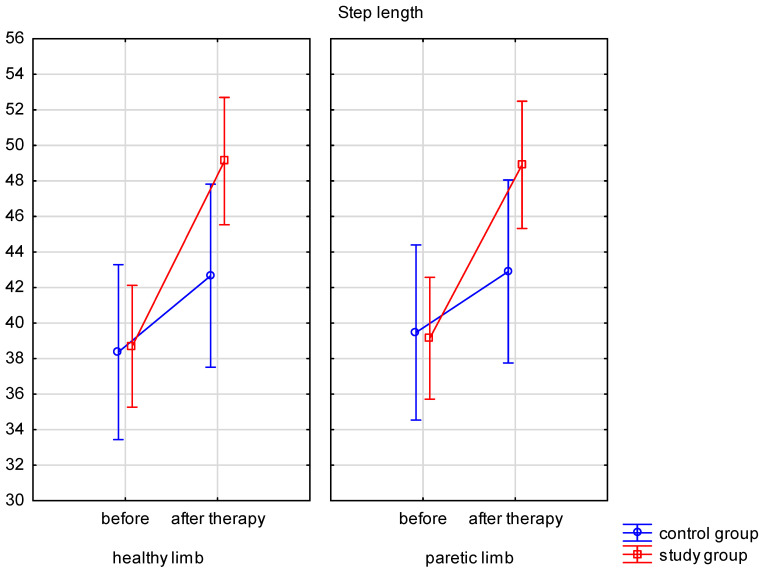
Step length before and after therapy.

**Figure 2 ijerph-19-16925-f002:**
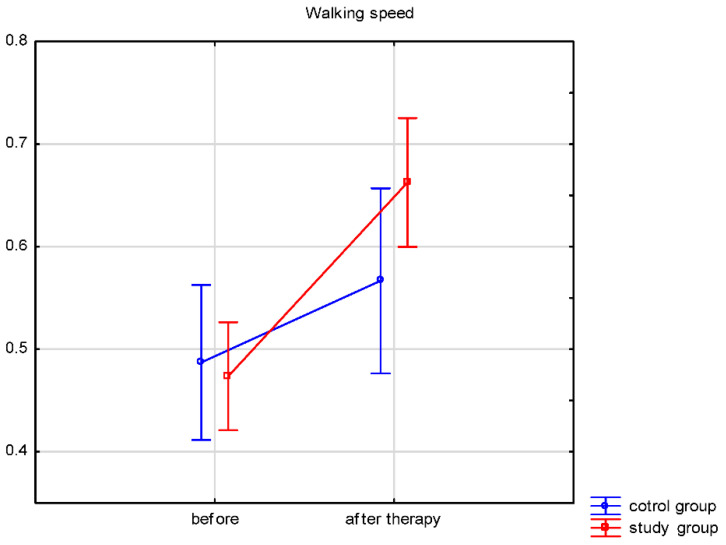
Walking speed before and after therapy.

**Figure 3 ijerph-19-16925-f003:**
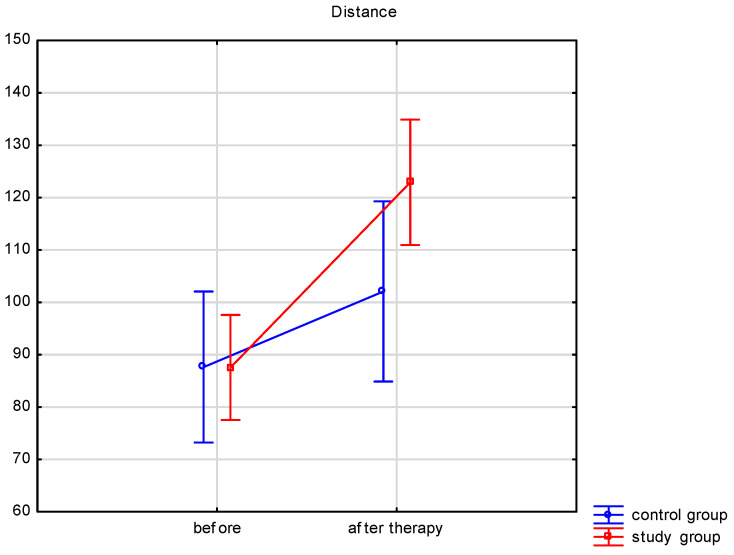
Distance before and after therapy.

**Table 1 ijerph-19-16925-t001:** Characteristics of the study population.

Variable	Study Group			Control Group			
	Mean ± SD	Median	Min–Max	Mean ± SD	Median	Min–Max	*p*
age	63.0 ± 10.9	65.0	27.0–85.0	61.9 ± 14.0	61.0	30.0–87.0	0.816
height (m)	1.69 ± 0.10	1.68	1.43–1.850	1.70 ± 0.014	1.70	1.43–2.10	0.705
weight (kg)	77.8 ± 14.8	78.0	53.0–110.0	77.5 ± 13.9	74.0	54.0–108.0	0.647
BMI	27.4 ± 4.7	26.8	18.3–42.4	26.9 ± 4.1	26.2	18.7–36.0	0.708

**Table 2 ijerph-19-16925-t002:** Step length and limb loading results for study and control groups before and after the therapy.

Variables			Paretic Limb			Healthy Limb				
			Study Group	Control Group	*p* between Groups	Study Group	Control Group	*p* between Groups	Main Effect *p*	Group × Time Interaction
Step length [cm]	Before	mean ± SD median range	39.1 ± 12.6 38.5 15.0–67.0	39.5 ± 15.0 36.5 13.0–70.0	0.230	38.7 ± 13.0 37.0 10.0–64.0	38.4 ± 15.8 35.5 7.0–68.0	0.916	<0.001	0.018
	After therapy	mean ± SD median range	48.9 ± 13.3 50.0 12.0–67.0	42.9 ± 15.8 45.0 12.0–72.0	0.056	49.1 ± 13.1 51.5 19.0–71.0	42.7 ± 17.0 43.0 9.0–75.0	0.040		
	*p **		<0.001	0.023		<0.001	0.005			
Limb loading (time on limb) [%]	Before	mean ± SD median range	49.5 ± 3.5 50.0 37.0–59.0	49.5 ± 3.0 49.5 39.0–54.0	0.913	50.4 ± 3.5 50.0 41.0–63.0	51.1 ± 3.7 50.0 44.0–72.0	0.709	0.219	0.722
	After therapy	mean ± SD median range	49.4 ± 4.0 50.0 28.0–65.0	49.4 ± 2.1 50.0 42.0–52.0	0.962	51.1 ± 3.7 50.0 44.0–72.0	50.7 ± 3.0 50.0 46.0–61.0	0.580		
	*p **		0.869	0.946		0.385	0.919			

* within-group effect.

**Table 3 ijerph-19-16925-t003:** Walking speed and distance traveled results for study and control groups prior to and following therapy.

Variables			Study Group	Control Group	*p* between Groups	Main Effect *p*	Group × Time Interaction
Walking speed [m/s]	Before	mean ± SD median range	0.47 ± 0.19 0.47 0.11–0.92	0.49 ± 0.25 0.47 0.14–1.31	0.642	<0.001	<0.001
	After therapy	mean ± SD median range	0.66 ± 0.24 0.63 0.19–1.25	0.57 ± 0.27 0.56 0.17–1.33	0.023		
	*p **		<0.001	<0.001			
Distance traveled [m]	Before	mean ± SD median range	87.5 ± 36.9 85.0 18.0–170.0	87.6 ± 45.2 84.0 25.0–237.0	0.768	<0.001	<0.001
	After therapy	mean ± SD median range	122.9 ± 47.3 116.5 34.5–238.0	102.1 ± 47.8 100.0 30.0–240.0	<0.001		
	*p **		<0.001	<0.001			

* within-group effect.

**Table 4 ijerph-19-16925-t004:** Functional gait variables.

Variable	Group	Before Therapy		After Therapy		
		Abnormal	Normal	Abnormal	Normal	*p*
Romberg test	study	47 (75.8%)	15 (24.2%)	20 (32.3%)	42 (67.7%)	<0.001
	control	19 (63.3%)	11 (36.7%)	8 (26.7%)	22 (73.3%)	0.006
Walking base	study	44 (71.0%)	18 (29.0%)	27 (43.5%)	35 (56.5%)	<0.001
	control	20 (66.7%)	10 (33.3%)	25 (83.3%)	5 (16.7%)	0.074
Orthopedic aid *	study	23 (37.1%)	39 (62.9%)	10 (16.1%)	52 (83.9%)	0.006
	control	21 (70.0%)	9 (30.0%)	10 (33.3%)	20 (66.7%)	0.006

* crutches/walker.

## Data Availability

The data presented in this study are available on request from the first author. The data are not publicly available due to ethical restrictions.

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
