# Peer review of "Effect of Treadmill Training with Visual Biofeedback on Selected Gait Parameters in Subacute Hemiparetic Stroke Patients"

_ijerph, 2022, doi:10.3390/ijerph192416925_

Round 1
Reviewer 1 Report
Thank you for the opportunity to review the manuscript (MS). The paper is interesting but unfortunately, needs some corrections, explanations, and additions. The authors should explain the group selection, statistical methods, and model design (2 independent variables). I have a few recommendations for the authors to address before publication.
My comments are as follows:
General comments:
1. The article should be written in the CONSORT scheme.
2. This is a clinical trial on patients - it should be registered.
Introduction
1. “The first includes muscle 59 strengthening exercises 22–24 and neuromuscular re-education (PNF, NDT-Bobath), which 60 increases ROM by increasing muscle length and improves neuromuscular performance.”à It is unclear, please define the ROM (range of motion) before using it.
2. “Only a few types of research have been dedicated to evaluating whether visual biofeedback plays an essential role in stroke recovery and may be a valuable approach to stroke rehabilitation 1,36”. àThe authors cite only 2 examples of similar studies. Perhaps it would be worthwhile to cite more available research on this subject.
3. Could authors specify what is new in this research, please?
4. The questions are not clear. Even in a simplified model, there should be "effects of the treadmill with feedback on..." in each question. How did the authors want to verify 1st question?
There is also no information about "feedback" in the second and third questions.
The second question - the authors did not study balance in gait, only Romberg.
Materials And Methods
1. In the first sentence of the methods there is information about random selection - Can the authors explain why the division into groups was unequal (CONSORT)? The basic split model is 1:1. If the authors used some other division it requires justification. I do not see any reason for an unequal division here. Can the authors explain why is the study group twice as large as the control group? What were the authors' guidelines for assigning participants to groups?
2. According to CONSORT, the method of draw should also be given in the article.
3. Was the sample size calculated (G Power analysis)?
4. We have two working factors here (two independent variables) - treadmill and visual feedback. So actually the experimental model should be built with more groups. There should also be a treadmill group without feedback and feedback without treadmill. Could the authors explain this?
5. “The gait parameters of each participant in the control and study groups were measured twice in the treadmill test. The first measurement occurred before the start of the treatment and again on the last day of treatment, after four weeks.” (lines 126-128)- Could the authors explain how the measurements looked like?
Procedure and Instruments:
1. A Romberg test was used to assess the balance which is used to diagnose gait disorders caused by incorrect proprioception with information about the location of the joints. Perhaps it would be better to use a posturograph to assess postural stability to obtain more reliable results. Computerized posturography is more sensitive than the clinical Romberg Test in detecting postural control impairment in evaluate patients with an impaired balance due to sensory and motor disorders. The authors also do not mention this obvious limitation in the limitations section
2. The description of the Romberg test should be supplemented with, for example, it is not clear from the description how to perform the Romberg test.
3. Were the results averaged over several trials? Taking into account generally low reliability of functional tests.
4. “Biodex Gait Trainer 3 treadmill with real-time step length visualization (where the 1foot was placed) was used to assess the selected spatiotemporal gait parameters (Figure 2).” (lines 166-167)"àPerhaps it would be worthwhile to present the experiment graphically for better understanding.
Statistical analysis:
1. Probably in these studies it is possible to build one statistical model using the two-way ANOVA - with the intergroup factor (study and control group) and the repeated measure factor- time (I term- II term).
2. ANOVA is robust for lack of normal distribution. Further, interaction and main effects (with effect size), and post-hoc analysis could be used (e.g. Bonferroni correction).
Results
1. Line 223: “tep length’’ typo mistake, please correct it.
2. The presentation of the data in tabular form does not make it easy to determine if there were any practical differences in the results. It is suggested that graphs are added. If there are practical results to be highlighted, it may make the results more clear.
Discussion
1. The discussion section should focus on interpretation of the results and their application in science or real world applications.
2. “Exercises used in our rehabilitation program to strengthen the muscles of the lower limbs, exercises for motor control of the paresis of the lower limb, and training for postural, static, and dynamic balance undoubtedly improve walking speed for people after a stroke.”àUnfortunately, we don't know how treadmill training improved motor function in stroke patients. The improvement of motor functions is the result of both rehab and treadmill training. There is a possibility that earlier rehabilitation training significantly affected the results. Perhaps in future research, these 2 types of training should be separated in order to verify the true effects of treadmill training. Could the authors explain this?
3. The study aimed to analyze the effects of a treadmill with a visual feedback rehabilitation program in early post-stroke physiological gait reeducation. However, in the research questions, the authors ask only about the results of treadmill training with visual feedback. No mention of the rehabilitation program. Moreover, in summary, and conclusion the authors write that standard rehabilitation supplemented by a treadmill with visual biofeedback training significantly improved walking parameters. So the summary does not answer the research questions. Perhaps the research questions should be formulated differently or reconstructed summary and conclusions.
Conclusions
1. Conclusions are not meaningful, the authors do not show quantitative results. The conclusions should answer the aims of the study.
2. In addition, while the practical applications are almost obvious, perhaps they should be mentioned.
Reviewer 2 Report
Dear Authors,
Congratulations for your work.
You tried to contribute to the knowledge about the possible effects of Treadmill Training With Visual Biofeedback on Selected Gait Parameters in Subacute Hemiparetic Stroke Patients. After reading your article I identified several things that should be made before being suitable for publication in IJERPH. Format the article according to IJERPH (e.g. References in brackets;) and improve the English in your article.
Here are my considerations:
Abstract:
Add subsection Background
Add introductory sentence about manuscript not only aim of the study in background
Add data of booth genders(eg. age, height, weight) in methods
Describe what the participants did in the training period.
Introduction
Line 44-45 – Add reference.
Line 46 – what publication?
Line 70-73 – If there is a common practice add references of studys.
Remove 1.1 Aim – Line 80
Rewrite Line 84
Materials and methods
Rewrite Line 116-122 – it´s not clear
Line 166 – Add manufacturer details
Remove figure 2.
Line 172-175- rewrite to a single sentence
Why you did not use ANOVA repeated measures?
Results
What is p1 in Line227 and p2 in Line 228
Put in italics the “p” of the significance
Discussion
Line 276 – change , for .
Line 277-281 – Its not clear what you want to express. Please rewrite.
Line 305-313 – Add references to support this.
Line 338-344 – Your article significance is about visual biofeedback and you only discussed in 6 lines. Strength your results with a more detailed discussion with other studies that used biofeedback also.
Change the limitations to the discussion section and add more future recommendations such size of the sample, duration of the training….
Conclusion
Line 353-357 – Rewrite to a single sentence and not by points
Round 2
Reviewer 1 Report
Dear Autors,
Unfortunately, the manuscript still needs some updating and corrections. The biggest issue is that studies are still not aligned with the CONSORT guidelines. The authors also omitted one of the major issues concerning the registration of the trial. This is a clinical trial on patients - it should be registered. Did the authors register the trial?
Please see my other comments of some points below:
INTRODUCTION:
Point 1: “The first includes muscle 59 strengthening exercises 22–24 and neuromuscular re-education (PNF, NDT-Bobath), which 60 increases ROM by increasing muscle length and improves neuromuscular performance.” It is unclear, please define the ROM (range of motion) before using it.
Response 1: Thank you for your advice. As you suggest, we define the ROM (Introduction section, page 2, line 61)
Comment on response 1: Please replace the capital letters in "OF" with lowercase letters.
Point 4: The questions are not clear. Even in a simplified model, there should be "effects of the treadmill with feedback on..." in each question. How did the authors want to verify 1st question? There is also no information about "feedback" in the second and third questions. The second question - the authors did not study balance in gait, only Romberg.
Response 4:As you suggest, we have corrected the questions to: 2 “We have formulated the following research areas:
1. Effects of the treadmill with feedback on the step length in gait therapy
2. Effects of the treadmill with feedback on improved functional gait variables, such as sense of balance and walking base, and decreasing the need for orthopedics aids
3. Effects of the treadmill with feedback on walking speed”
And the answer is the result of the comparison of the measured values (e.g question 1. step length) before and after four weeks of therapy
Comment on response 4:
-Please add “visual” to the questions e.g. “Effects of the treadmill with visual feedback on the step length in gait therapy.”
-The reviewer did not expect an obvious answer that outcome was measured. The question is how to distinguish between these 2 factors (please see point 8).
MATERIALS AND METHODS
Point 5: In the first sentence of the methods there is information about random selection - Can the authors explain why the division into groups was unequal (CONSORT)? The basic split model is 1:1. If the authors used some other division it requires justification. I do not see any reason for an unequal division here. Can the authors explain why is the study group twice as large as the control group? What were the authors' guidelines for assigning participants to groups?
Response 5: Please consider that we are a scientific unit and a contractor providing rehabilitation services for the Wielkopolska region. The treadmill with biofeedback was our new purchase of rehabilitation equipment. Initially, the groups were of equal size. However, to comprehensively assess the treadmills with biofeedback therapy - our new equipment, we decided to let more people use biofeedback therapy on the treadmill. We were guided by the belief that a more extensive study sample size would affect the reliability of the results. Based on the data available in the literature (Carmen M. Cirstea. Gait Rehabilitation After Stroke Should We Re-Evaluate Our Practice? Stroke 2020;51(10), 2892-2894; doi:10.1161/strokeaha.120.032041 ), our control group confirms the already-known results. The common element of therapy for each group was traditional rehabilitation, and the treadmill with biofeedback was the only factor that differed between the compared gait re-education programs.
Comment on response 5: As the authors wrote initially, the groups were of equal size. However, in order to comprehensively assess treadmills with biofeedback therapy - the authors decided to increase the number of people using biofeedback therapy on the treadmill. If this was indeed the case and the group was increased during the project, it cannot be written that the trial was randomized. This information must be removed from the text as well as from the chart.
Point 6: According to CONSORT, the method of draw should also be given in the article.
Response 6: Initially, the groups were of equal size, and the participants were randomly distributed. However, to evaluate the effectiveness of the treadmill with biofeedback therapy - our new equipment, we decided to train on a treadmill with biofeedback in the double study group.
Comment on response 6: As already written above in point 5, it cannot be assigned as a random trial. Therefore authors are also unable to provide a draw method. This matter should be clarified in the manuscript.
Point 8: We have two working factors here (two independent variables) - treadmill and visual feedback. So actually the experimental model should be built with more groups. There should also be a treadmill group without feedback and feedback without treadmill. Could the authors explain this?
Response 8: The Biodex Gait Trainer automatically turns on the biofeedback screen during training. Unfortunately, the research instruments at our Clinic's disposal did not allow for the proposed division into groups. Thank you for your attention, and we may expand the research in line with your guidance in the future.
Comment on response 8: The reviewer was referring to the problem with methodology, not equipment. Could the authors refer to this?
PROCEDURE AND INSTRUMENTS:
Point 12: Were the results averaged over several trials? Taking into account generally low reliability of functional tests.
Response 12: In accordance with the rules of research carried out in our clinic, the test was carried out 3 times and the given result is the averaged value.
Comment on response 12: Is this information included in the manuscript? If not, please add it.
STATISTICAL ANALYSIS:
Point 17: The presentation of the data in tabular form does not make it easy to determine if there were any practical differences in the results. It is suggested that graphs are added. If there are practical results to be highlighted, it may make the results more clear.
Response 17: As suggested, we have added graphs to show the effects of both treatments better. Graphical changes in step length and limb loading are present in Figures 3 and 4. (Result section, line 239). Changes in walking speed are illustrated in Figure 5 (Result section, line 250)
Comment on response 17:
-The analysis is done for the paretic and healthy limb. Unfortunately, this is not included in the research questions.
- What are the primary (e.g. step length) and secondary outcomes - it should be precisely defined according to CONSORT. This information should be also included in the study.
- Figure 3 should be changed analogously to figure 4. Please correct it.
Author Response
Dear Reviewer,
Thank you very much for your time and effort in reading our article. I have attached replies to comments and proposed changes.
However, at the moment, in such a short time, we cannot refine all suggestions. If the paper is not accepted, we will regretfully have to withdraw the article from IJERPH. We then complete these studies, including a suggested trial registration. Once this process is complete, we'll resend the revised report.
Thanks again for your work.
Katarzyna Kaźmierczak

Reviewer 2 Report
Dear authors,
you decided not to accept almost all my suggestions to improve your article.
I do not agree with several parts of your article:
1.Abstract do not show what you did in your work. Where are the two groups? where are the charateristics of booth groups? What did each group?
2.Introduction: Add websites in the text? where is the reference of "Normative values for a healthy fast, walking adult range between 0,82 and 48 1,62 m/s. "?
3.Methods: Manufacturer details where are they? where was made? Anova of repeated measures should be done (Two groups and several variables). Remove figure 2, does not make sense.
4. Results: P still not in italics
5.Discussion: Discussion stills weak. Limitations of the study continues to be after the conclusions.
6.Conclusions are not topics.
Your papper is not formated according to IJERPH guidelines.
Author Response
Dear Reviewer,
Thank you very much for your time and effort in reading our article. I have attached replies to comments and proposed changes.
However, at the moment, in such a short time, we cannot refine all suggestions. If the paper is not accepted, we will regretfully have to withdraw the article from IJERPH. We then complete these studies, including a suggested trial registration. Once this process is complete, we'll resend the revised report.
Thanks again for your work.
